# Preparation of High-Efficiency Flame-Retardant and Superhydrophobic Cotton Fabric by a Multi-Step Dipping

**Jingda Huang** [1,†]**, Mengmeng Li** [1,†]**, Changying Ren** [1]**, Wentao Huang** [1]**, Qiang Wu** [1]**, Qian Li** [1]**, Wenbiao Zhang** [1] **and Siqun Wang** [1,2,*]

1   School of Chemistry and Materials Engineering, Zhejiang A & F University, Hangzhou 311300, China; hjd1015@163.com (J.H.); Limm0280@163.com (M.L.); rcydyx@163.com (C.R.); h0106w@163.com (W.H.); wuqiang@zafu.edu.cn (Q.W.); liqian_polymer@126.com (Q.L.); zwb@zafu.edu.cn (W.Z.)
2   Center for Renewable Carbon, University of Tennessee, Knoxville, TN 37996, USA
*   Correspondence: swang@utk.edu
†   The authors contributed equally to the article.

**Abstract:** Cotton fabric, as an important material, is suffering from some defects such as flammability, easy pollution and so on; therefore, it is important to make a flame-retardant and superhydrophobic modification on cotton fabric. In this study, we demonstrated a preparation of high-efficiency flame-retardant and superhydrophobic cotton fabric with double coated construction by a simple multi-step dipping. First, the fabric was immersed in branched poly(ethylenimine) (BPEI) and ammonium polyphosphate (APP) water dispersions successively, and then immersed in polydimethylsiloxane (PDMS)/cellulose nanocrystals (CNC)-SiO$_2$ toluene dispersion to form a BPEI/APP/PDMS/CNC-SiO$_2$ (BAPC) composite coating on the surface of the cotton fabric. Here, the hydrophobic modified CNC-SiO$_2$ rods were used to construct the superhydrophobic layer and the BPEI/APP mixture was used as the flame-retardant layer, as well as SiO$_2$ particles which could further improve the flame-retardant effect. PDMS was mainly used as an adhesive between the BPEI/APP layer and the CNC-SiO$_2$ layer. The resulting cotton fabric shows outstanding flame-retardant properties, in that the value of oxygen index meter (LOI) reaches 69.8, as well as excellent superhydrophobicity, in that the water contact angle (WCA) is up to 156.6°. Meanwhile, there is a good abrasion resistance, the superhydrophobicity is not lost until the 16th abrasion cycles and the flame retardant retains well, even after 100 abrasion cycles in an automatic vertical flammability cabinet under a pressure of 8.8 kPa.

**Keywords:** cotton fabric; flame retardant; superhydrophobic; immersing

## 1. Introduction

Fire disaster occasionally takes placed in daily life and may lead to significant losses of human lives and properties [1]. Additionally, flame-retardant materials could repel or slow the spread of fire and have been widely used [2]. Cotton fabric is a common and important material for human survival due to its softness, comfort and environmental protection, but it is also too flammable to use in some occasions such as use as for fabric for fireman's clothes, which have a high requirement for flame-retardancy [3]. Therefore, it is indispensable to develop a high flame retardancy cotton fabric. To date, there have been various methods exploited to prepare a flame-retardant cotton fabric, including dipping [4], pad–dry–cure [5], sol–gel [6], layer-by-layer assembly [7] and so on. The common purpose of these methods is mainly to stabilize fire retardant on the surface of the cotton fibers. Of all of them, dipping is a mature and efficient method due to its convenient operation and appropriate large-scale production [8]. At present, there are various types of fire retardants, commonly including halogenated flame retardants, phosphorus flame re-

tardants, nitrogen flame retardants and inorganic flame retardants [9]. From an environmental view, phosphorus flame retardants are targeted for further development because of the advantages of low smoke, non-toxic and low halogen. For example, Kanat et al. [10] synthesized two new phosphorus flame retardants for cotton fabric using an oxide as raw material, and their flame-retardant effect was obvious. Phosphorus flame retardants were also often used in conjunction with other flame retardants. For example, Alongi et al. [11] applied both $SiO_2$ and phosphorus flame retardant to cotton fabric by sol–gel method, which greatly enhanced the flame-retardant effect.

At present, the development of multifunctional materials, such as magnetic adsorption [12], fluorescent adsorption [13], flame retardant-conductive [14] materials, etc., is a trend due to the social demand for high value-added products [15]. Additionally, a superhydrophobic surface which requires reasonable rough structure and low surface free energy has such functions as self-cleaning, antifouling and resistance reduction, etc., attracting scholars' interest [16,17]. Therefore, it is a feasible and good choice to add both flame-retardant and superhydrophobic functions to cotton fabric to make it both waterproof and fire retardant. For example, Xue et al. [18] took synthetic polyacrylate soap-free latex as a hydrophobic modifier and APP as flame retardant to form a composite mixture, and a superhydrophobic and flame-retardant cotton could be prepared only by a multi-step immersing. Additionally, a self-healing and flame-retardant superhydrophobic surface can also be achieved in cotton using branched poly(ethylenimine) (BPEI) and ammonium polyphosphate (APP) as promoters by immersing in turn, followed by treating with fluorinated-decyl polyhedral oligomeric silsesquioxane, which was synthesized by perfluorodecanethiol and vinyl-decyl polyhedral oligomeric silsesquioxane, and the resulting cotton also showed excellent self-healing properties [19]. All the above indicated that the preparation of the flame-retardant and hydrophobic cotton fabric has made some progress.

However, there is still room for improvement in flame retardancy of cotton fabric, and superhydrophobic surfaces are also suffering from poor mechanical properties [17]. In particular, it is tougher to fix a superhydrophobic coating on the surface of substrates with flame-retardant treatment. For example, parts of superhydrophobic coatings have insufficient adhesion and are easy to disengage [20], or even if the adhesion is sufficient, the rough structure is prone to collapse [21]. In view of the above problems, our research group used cellulose nanocrystals (CNC) as skeletons to string $SiO_2$ particles together to consolidate a rough structure on the surface of a commercial spray adhesive. Compared with the pure $SiO_2$ one, the resulting CNC-$SiO_2$ superhydrophobic coating showed a significant improvement on abrasion resistance [22].

From the above points, it is clear that during the current development, it is difficult for a flame retardant and superhydrophobic cotton fabric to meet a wider application requirement. Therefore, the objective of this study was to enhance flame retardancy of cotton fabric and to improve stability of superhydrophobic coatings on it. Both BPEI and APP were adopted to build a flame-retardant layer. The 1H,1H,2H,2H-perfluorooctyltriethoxysilane (FOTS)-modified CNC-$SiO_2$ rods were used to build a superhydrophobic layer and also beneficial to enhance the flame retardant performance. Because $SiO_2$ was an inorganic flame retardant material and could construct a dual flame retardant structure with BPEI/APP. PDMS is used as an adhesive which could provide adhesion between the BPEI/APP and the FOTS-modified CNC-$SiO_2$ rods to prepare a highly flame-retardant and superhydrophobic cotton fabric with good stability. In this study, the used substances such as BPEI, APP, PDMS and CNC-$SiO_2$ are considered non-toxic, and the C–F bond is stable in FOTS. However, they should be used with caution.

## 2. Materials and Methods

### 2.1. Materials

Branched poly(ethylenimine) (BPEI), ammonium polyphosphate (APP), 1H,1H,2H,2H-perfluorooctyltriethoxysilane (FOTS, $C_8F_{13}H_4Si(OCH_2CH_3)_3$, 97%) and tetraethyl orthosilicate (TEOS, 98%) were purchased from Sigma (Shanghai, China). The cellulose nanocrystals (CNC) with solid of 12% were purchased from the Process Development Center, University of Maine (Orono, ME, USA). Ammonium hydroxide (25%), toluene (99.9%) and anhydrous ethanol (99.5%) were purchased from Macklin (Shanghai, China). Polydimethylsiloxane (PDMS, Sylgard 184 silicone elastomer) and its curing agent were purchased from Dow Corning, Inc. (Midland, MI, USA). The cotton fabric, made of pure cotton, was purchased from Yilian Jiamei Household Products Co., Ltd., Beijing, China, and washed with anhydrous ethanol and air dried before using. All the chemical reagents were used without any treatment.

### 2.2. Preparation of Flame-Retardant Cotton Fabric

Similar to the description in this study [19]. First, BPEI aqueous solution (5 mg/mL), APP aqueous dispersion (20 mg/mL) and PDMS (mass ratio of PDMS to curing agent = 10:1) toluene solution (2 mg/mL) were respectively modulated for standby application. The cotton fabric washed with anhydrous ethanol and dried in air was entirely immersed in the BPEI aqueous solution for 30 min and wrung out, followed by immersion in APP dispersion aqueous for 90 min and then washed by deionized water. Subsequently, the BPEI/APP (BA) cotton fabric was dried at 80 °C until the water was completely evaporated. Finally, the dried BA cotton fabric was immersed in PDMS toluene solution for 10 min.

### 2.3. Preparation of Hydrophobic CNC-SiO₂ Rods

The hydrophobic CNC-SiO₂ rods were taken as structural materials for superhydrophobic coating and could be fabricated as described in our previous study [22]. Briefly, 50 mL of CNC ethanol suspension (1.5 wt.%) were mixed with 4 mL of TEOS under alkaline conditions (pH = 13) and 55 °C for 2 h to obtain the CNC-SiO₂ rods. The hydrophobic CNC-SiO₂ rods could be prepared by adding FOTS and stirring for another 2 h. To remove unreacted FOTS and redundant by-product, the hydrophobic CNC-SiO₂ rods were washed using fresh toluene by centrifugation.

### 2.4. Preparation of Flame-Retardant and Superhydrophobic Cotton Fabric

The hydrophobic CNC-SiO₂ rods were added to PDMS (including curing agent) toluene solution and stirred for 20 min to obtain the PDMS/CNC-SiO₂ hydrophobic mixture (10 wt.%). The above cotton fabric treated with BPEI/APP/PDMS (BAP) was immersed in the PDMS/CNC-SiO₂ hydrophobic mixture for 20 min and air dried, followed by drying at 120 °C for 2 h to obtain the flame-retardant and superhydrophobic cotton fabric (BPEI/APP/PDMS/CNC-SiO₂ (BAPC) cotton fabric). To make a comparison, the samples including BPEI cotton (B cotton), BPEI/APP cotton (BA cotton) and BPEI/APP/PDMS cotton (BAP cotton) are prepared by the same method.

### 2.5. Flammability and Antiabrasion of Cotton Fabric

As described in the study [19], the flammability of the samples with different treatments was estimated by a vertical flame test with a size of 8 cm × 30 cm using an automatic vertical flammability cabinet (5402, Vouch testing technology Co., Ltd., Suzhou, China). The samples were kept in fire from a gas burner for 12 s and then removed. The mechanical property of the coated cotton fabric was tested by an antiabrasion test using a commercial fabric abrasion tester (Y571, Laizhou Yuanmao Instrument Co. Ltd., Laizhou, China) with a speed of 28 cm/s under an applied pressure of 8.8 Kpa.

*2.6 Characterization*

A Zeiss Auriga SEM/FIB crossbeam workstation (Zeiss, Oberkochen, Germany) was used to collect STEM images. SEM images were observed by a QUANTA FEG 250 (FEI company, Hillsboro, OR, USA). Chemical construction of the samples with different treatment was analyzed by a FTIR analyzer (PerkinElmer, Waltham, MA, USA) with a range from 600 to 4000 cm$^{-1}$ with scans of 16 times. DTG-60 thermogravimetric analyzer (Shimadzu, Kyoto, Japan) was used to analyze thermostability of APP, CNC-SiO$_2$ rods before and after modification. An X-ray photoelectron spec-troscopy (XPS, PHI-5300 photoelectron spectrometer, PerkinElmer Instruments Co. Ltd., Waltham, MA, USA) was used to determine elemental composition of the cotton fabric before and after treatment. An X-ray diffractometer (XRD-6000 Shimadzu, Kyoto, Japan) was used to analyze the change of crystallinity of CNC-SiO$_2$ rods and elemental information of other samples. Both water contact angle (WCA) and slide angle (SA) were measured using a commercial contact angle meter (Shanghai Zhongchen JC2000D, Shanghai, China). An oxygen Index Meter (LOI, JF-3, Jiangning District Analytical Instrument Factory, Nanjing, China) was used to analyze flame-retardant property.

## 3. Results and Discussion

### 3.1. Formation Mechanism of Flame-Retardant and Superhydrophobic Cotton Fabric

As shown in Figure 1, the washed cotton fabric was successively immersed in the BPEI and APP aqueous suspension to obtain the APP/BPEI flame-retardant coating on the cotton fabric [23]. In this flame-retardant system, the APP [24,25] acted as the acid source and a stable polyphosphoric acid could be generated and play a role on protection of the polymer and isolate oxygen when APP was decomposed by heating [26]. The BPEI acted as a blowing agent and carbon source and occurs thermal decomposition under the catalysis of APP to produce loose carbon layer due to its decomposition and the production of typical gases, meanwhile supporting adhesion between the APP particles and the cotton fabric due to its high viscosity [19], followed by immersing in the PDMS to make the APP firmer on the fiber surface. Finally, PDMS/CNC-SiO$_2$ hydrophobic mixture was used to cover the BAP cotton and the modified CNC-SiO$_2$ rods formed a superhydrophobic surface on the top of the BPEI/APP layer by meeting both reasonable rough structure and low surface free energy supplied by FOTS. At the same time, the SiO$_2$ was also an inorganic flame-retardant matter [27] and could further enhance the flame-retardant effect. Additionally, PDMS could supply the adhesion between the CNC-SiO$_2$ layer and the BPEI/APP layer. Therefore, the high-efficiency flame-retardant and superhydrophobic cotton fabric (BAPC cotton) could be prepared.

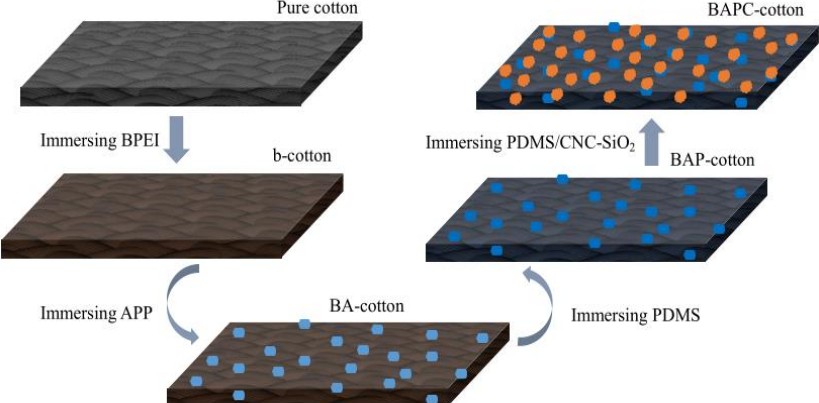

**Figure 1.** Formation mechanism of flame-retardant and superhydrophobic cotton fabric.

## 3.2. Surface Morphology

TEM images of the CNC-SiO₂ rods can be seen in Figure 2a. CNC is a substance rod-like structure, and the pearl-like CNC-SiO₂ rods are formed by growing SiO₂ particles on the CNC as a central axis. As shown in Figure 2b, the cotton fibers showed a relatively smooth surface in the untreated fabric and have a certain of gaps among them, which are the reason for good breathability of the cotton fabric. After the fabrics were coated by BPEI (Figure 2c), there was no significant change on the surface of the cotton fibers. After further immersing in APP aqueous dispersion, some irregular-shaped APP particles were attached to the surface of the B cotton fabric (Figure 2d), followed by treatment by PDMS (Figure 2e) to obtain a firmer structure. When the CNC-SiO₂/PDMS hydrophobic mixture was immersed onto the BA cotton, the flocked CNC-SiO₂ rods were fixed to the APP/BPEI layer using PDMS as an adhesive to build a stable rough structure (as shown in Figure 2f); the low surface free energy is supplied by the modifier FOTS. The flame-retardant and superhydrophobic BAPC cotton fabric is obtained.

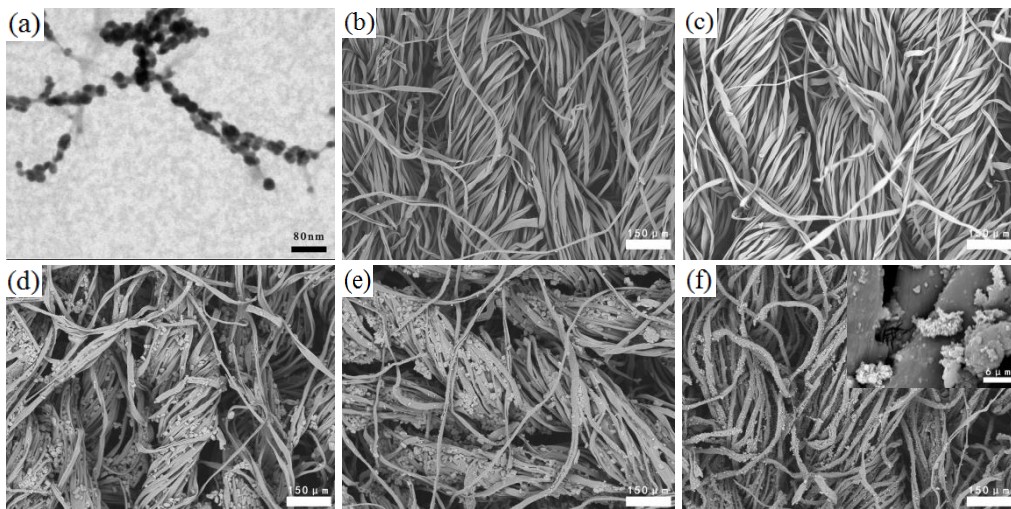

**Figure 2.** STEM image of (**a**) CNC-SiO₂ rods; SEM images of the (**b**) pure cotton, (**c**) B cotton, (**d**) BA cotton, (**e**) BAP cotton, and (**f**) BAPC cotton including high magnification inset.

## 3.3. TGA of Cotton Fabric with Different Treatment

The thermal stability of the cotton fabric with different treatments was tested by a TG analyzer. As shown in Figure 3, this was in the temperature range of 25–100 °C [28]. It mainly occurs during the evaporation of absorbed water; during 100–200 °C, the weight loss rates of all the samples were relatively stable. The cotton fibers decomposed into carbon by removing hydroxyl groups during 200–400 °C, while in a higher temperature range of 400–600 °C, the residues were slowly decomposed into volatile products containing levoguloglucan [29]. In the temperature range of 400–600 °C, in the B cotton, in addition to the carbonization of the cotton fibers, BPEI also produced bubbles by thermal decomposition and formed loose carbon layer on the surface of the cotton fibers. Therefore, the residue mass is greater than the pure one. The mass of the BA cotton is mainly rapidly deteriorating due to the thermal decomposition of APP during 200–300 °C [30]. The BAP cotton shows an almost coincidental thermal stability curve with the BA one, because the amount of PDMS is extremely low and has little effect on the residual mass. In the BAPC cotton, in addition to the thermal decomposition of cotton fibers and CNCs of the CNC-SiO₂ rods during 200–400 °C, the dehydration reaction of hydroxyl groups among SiO₂ particles is also one of the main reasons [31]. In the range of 400–600 °C, SiO₂ continues to dehydrate and lose weight. Additionally, the weight loss of the modified CNC-SiO₂ rods in the range of 400–600 °C might be caused by the thermal decomposition of its surface low-energy substances, and the mass is relatively stable after 600 °C.

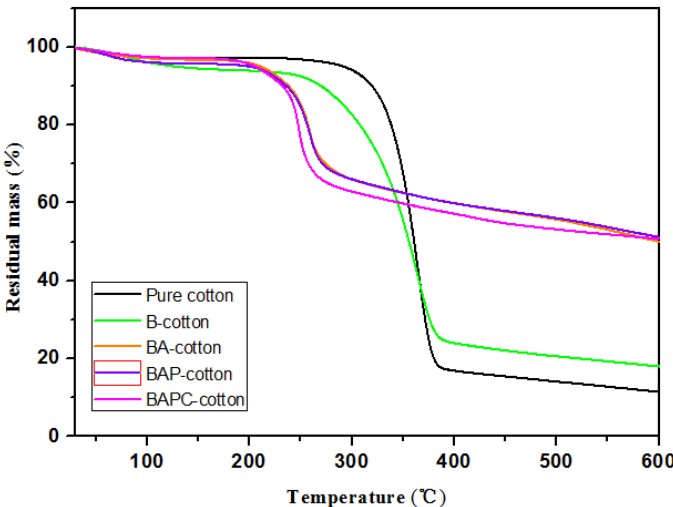

**Figure 3.** TGA curves of the pure, B, BA, BAP and BAPC cotton fabric.

*3.4. FTIR Analysis*

As shown in Figure 4, the absorption peak at 1100 cm$^{-1}$ is caused by the stretching vibration of the Si–O–Si bond from $SiO_2$ particles [32]. The absorption peak at 2988 cm$^{-1}$ is due to the stretching vibration of the C–H bonds from CNC and FOTS [33]. The stretching vibration of the –OH causes the absorption peak at 3439 cm$^{-1}$ and the strength of the modified one is weaker than that of the unmodified one. This is because –OH is reduced due to the reaction between FOTS and –OH on the $SiO_2$ surface, the reaction between FOTS and –OH has generated new Si-O bonds whose stretching vibration caused the new absorption peak at 901 cm$^{-1}$ [16]. The new absorption peaks at 1145 and 1208 cm$^{-1}$ are caused by the stretching vibration of the C–F bond from FOTS, proving successful grafting of FOTS.

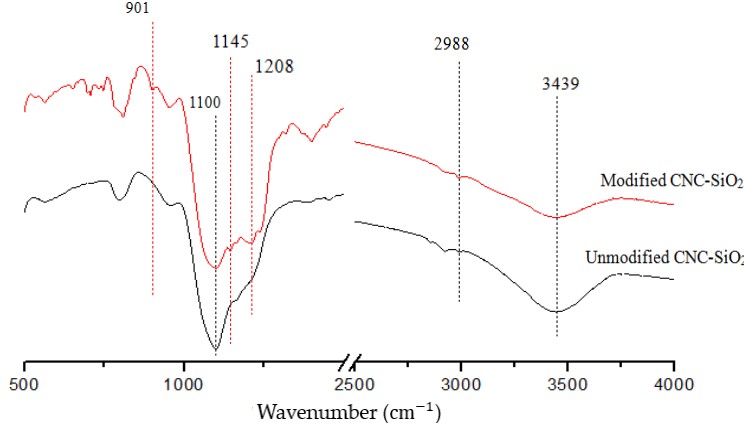

**Figure 4.** FTIR of CNC-$SiO_2$ rods before and after modification.

*3.5. Mapping Analysis*

To detect the chemical component of the BAPC cotton fabric, energy dispersive spectrometer (EDS) is used. As shown in Figure 5a–h, except C and O elements, the elements (N, P and Si) exist in the BAPC cotton and their distribution is relatively uniform, proving that APP, BPEI and PDMS have been attached to the surface of the cotton fiber with good distribution. After treatment with the modified CNC-$SiO_2$ rods, the fabric shows a dense distribution of the F element, which supplies low surface free energy for the superhydrophobic surface. It has been proven that FOTS are grafted onto CNC-$SiO_2$ rods in FTIR analysis (Figure 4).

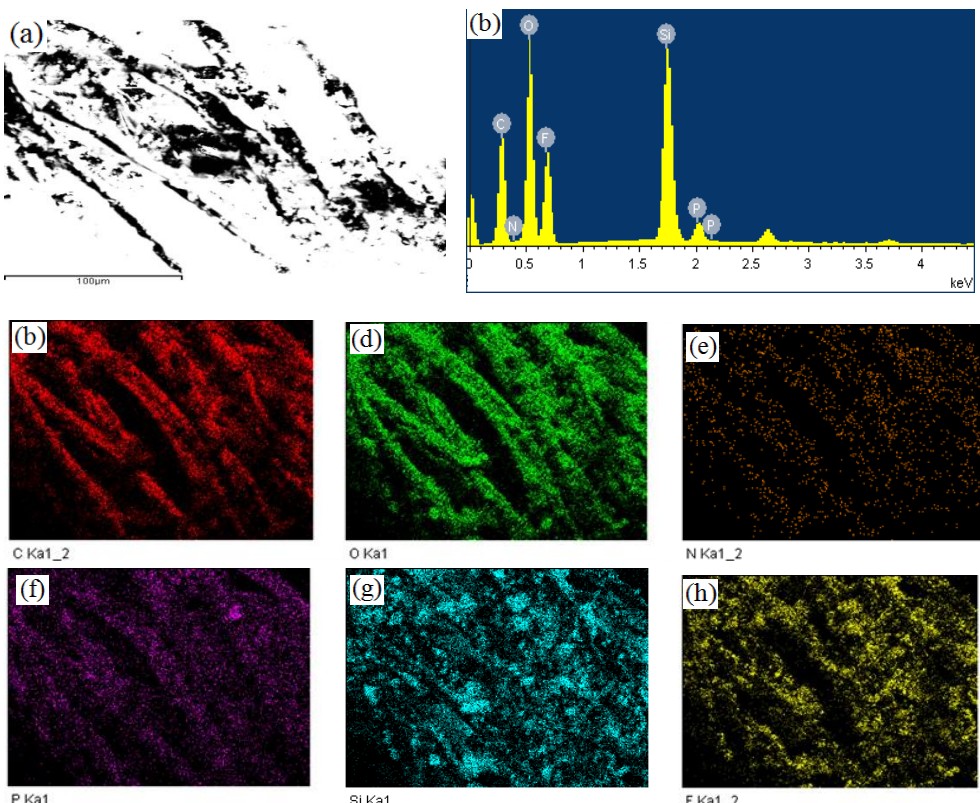

**Figure 5.** EDS spectrum of the BAPC cotton fabric: (**a**) SEM image of the area, (**b**) general elemental map and the corresponding element distribution of (**c**) C, (**d**) O, (**e**) N, (**f**) P, (**g**) Si and (**h**) F.

*3.6. Wettability*

As shown in Figure 6a–c, all the WCAs of the pure, B and BA cotton are 0°, and when the water droplets (dyed blue) were dropped onto the surface of these cottons, they could soak into the substrates due to gaps among cotton fibers and a large number of hydrophilic hydroxyl groups on the surface of these cotton fibers. When it was continued to be treated only by PDMS (Figure 6d), the sample showed a certain hydrophobicity (WCA = 127.8°), because PDMS itself is a hydrophobic material and could provide a certain low surface free energy. After the treatment with the modified CNC-SiO$_2$ rods (Figure 6e), the WCA was up to 156.6° and the SA only 7°, proving that modified CNC-SiO$_2$ rods could provide sufficient roughness and low surface free energy [22].

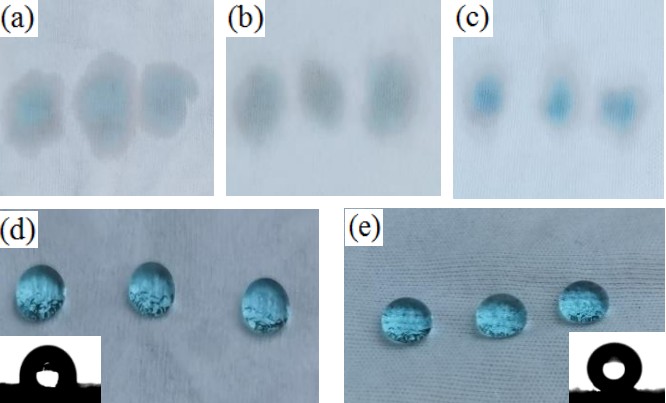

**Figure 6.** Pictures of water droplets on the surfaces of the (**a**) pure cotton fabric, (**b**) B cotton fabric, (**c**) BA cotton fabric, (**d**) BAP cotton fabric (including its droplet image) and (**e**) BAPC cotton fabric (including its droplet image).

### 3.7. Flame-Retardant Properties

A vertical flame test is conducted to evaluate flame-retardant properties of the cotton fabric with different treatment. As shown in Figure 7a, the pure cotton could be quickly lit by and the fire spreads fast. Additionally, when the flame is removed, the fire does not go out until it is absolutely burned. Additionally, the pure cotton is burnt to ashes, extinguishing the fire within 18 s. Additionally, the BA cotton was not ignited within 12 s; when the fire source was removed, the flame was extinguished instantly and the cotton was intact, only leaving a long blackened trace by smoke. This is because BPEI is dehydrated under the catalysis of APP during combustion and the foaming agent decomposes out inert gas to form an expansive and porous carbon layer, hindering the extension of oxygen and heat and playing a role in protecting the underlying cotton fabric [34]. The BAP cotton showed a similar combustion process to the BA one, but a much shorter blackened trace. This might be because after PDMS burned, a carbon layer was added and had a better protective effect. Similar to the BA and BAP ones, the flame was extinguished as soon as it was removed; the BAPC cotton could also not be ignited within 12 s and had the shortest blackened trace, showing outstanding flame-retardant properties. This is because $SiO_2$ itself is a good inorganic flame- retardant, combining together with BA coating and having a double protection [35].

As we know, the LOI value represents the minimum oxygen concentration required for the sample to maintain combustion in oxygen–nitrogen mixture. If the LOI value of materials is higher than 21%, it will be considered flammable. As shown in Figure 7b, the LOI value of the untreated sample is only 18.3% and consistent with the previous report which the LOI value of the pure cotton is only 18–19% [7]. The LOI value of the BA coating reaches 22% and belongs to a non-combustible material, proving that BA coating has a good flame-retardant effect. The LOI value of the BAP coating is 24.2% and a little higher than that of the AB one. This is because PDMS is an easily combustible material and burnt to form a loose carbon layer which plays a role on heat insulation and oxygen isolation. The LOI value of the BAPC coating is up to 68.6%, which is because $SiO_2$ itself is a kind of flammable material and able to further improve flame-retardant properties.

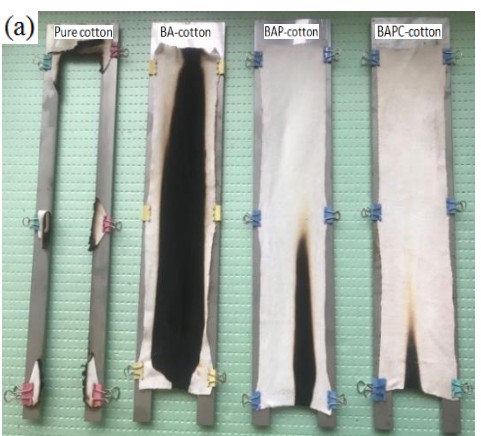
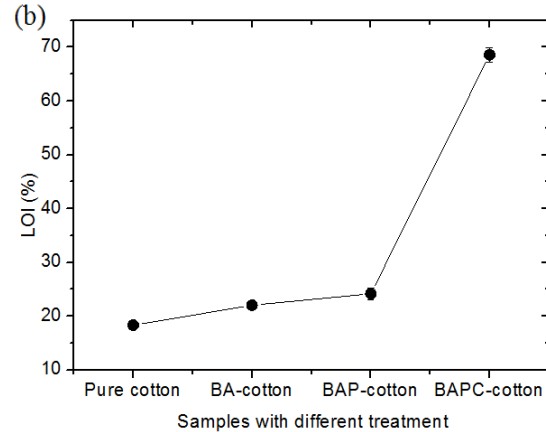

**Figure 7.** (**a**) Digital images of the cotton fabric with different treatment after vertical flame testing and (**b**) LOI curve of samples with different treatment.

### 3.8. Durability

To test the mechanical durability of the samples, the cotton fabric with different treatment was examined by repeatedly rubbing using a cylindrical copper under a pressure of 8.8 kPa. The BAPC cotton lost superhydrophobic until the 16th cycles and the WCA decreased to 149.4° (as shown in Figure 8a). This was because the CNC-$SiO_2$ coating on the surface of the cotton fiber was destroyed continuously during testing, leaving only sparse CNC-$SiO_2$ clumps and not affording superhydrophobic (as shown in Figure 8b). To prove

the flame-retardant effect of the BA coating, the cotton treated only with CNC-SiO₂ coating was taken as a reference group. After abrasion for the same 16 times, the vertical flame test was carried out on it. The results are shown in Figure 8c; the flames could spread quickly and the CNC-SiO₂ cotton was quickly burned out, which is because when the CNC-SiO₂ coating is worn off, it loses its fire protection. However, the flame-retardant performance of the BAPC coating was still retained even after one hundred cycles (as shown in Figure 8d), which is because the BA coating has not been worn off after 100 abrasion cycles. The SEM image of the BAPC coating after combustion is shown in Figure 8e. The fiber is intact, with only a loose surface layer, which is a porous carbon layer formed by the thermal decomposition of APP. This proves the BA coating has excellent adhesion. It still had a good flame-retardant effect with a LOI value of 22.4%. Therefore, it is proven that it has good flame-retardant durability.

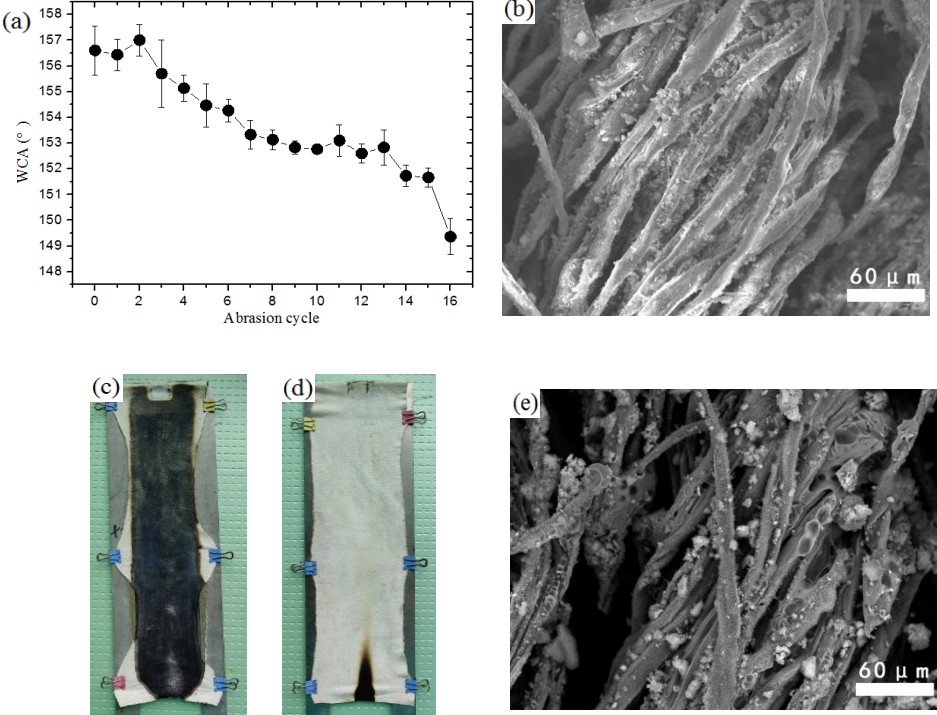

**Figure 8.** (**a**) Change of WCA of the BAPC cotton fabric with abrasion cycles, (**b**) SEM image of BAPC cotton fabric after abrasion, burning figures of (**c**) PC and (**d**) BAPC cotton fabric after abrasion and (**e**) SEM image of and burnt BAPC cotton fabric after abrasion.

## 4. Conclusions

The flame-retardant and superhydrophobic cotton fabric could be prepared by a simple sequential dipping. The flame-retardant component BPEI/APP and the superhydrophobic component CNC-SiO₂ rods could be connected together by PDMS to form a multifunctional coating on the surface of cotton fabric. This as-prepared cotton fabric showed excellent flame-retardant and superhydrophobic properties, the value of oxygen index meter (LOI) reaching 69.8 and WCA is up to 156.6°. In the BPEI/APP coating, BPEI acts as acid source and a binder and APP acts as carbon source and a foaming agent. During burning, a loose carbon layer is formed on the surface of the cotton fibers to isolate heat and oxygen and to play a flame-retardant role. PDMS mainly adheres the CNC-SiO₂ layer to the surface of BPEI/APP layer. In the CNC-SiO₂ layer, the surface of SiO₂ is hydrophobically modified by FOTS to provide rough structure and low surface free energy, while CNC mainly plays a role in stabilizing SiO₂ particles; meanwhile, SiO₂ is a highly effective flame retardant material. The BAPC cotton fabric also shows good wear resistance. After the 16th abrasion cycles, the sample lost its superhydrophobic properties, but still had

flame-retardant properties. Given the simplicity of preparation and good performance, the resulting cotton fabric has a potential application.

**Author Contributions:** Conceptualization, J.H.; Data curation, W.H.; Formal analysis, J.H.; Investigation, C.R. and Q.W.; Methodology, Q.L. and W.Z.; Resources, S.W.; Writing–original draft, M.L. All authors have read and agreed to the published version of the manuscript.

**Funding:** This work was supported by the National Natural Science Foundation of China (Grant No. 31901246), Natural Science Foundation of Zhejiang Province (No. LY21C160002), Scientific Research Development Foundation of Zhejiang A & F University (Grant No. 2018FR054).

**Institutional Review Board Statement:** Not applicable.

**Informed Consent Statement:** Not applicable.

**Data Availability Statement:** The data presented in this study are available on request from the corresponding author.

**Acknowledgments**: The authors would like to thank national engineering and technology research center of wood-based resources provision and comprehensive use of an experimental platform in this experiment.

**Conflicts of Interest:** We declare that we do not have any commercial or associative interests that represent a conflict of interest in connection with the work submitted.

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
