# Peer review of "Preparation of High-Efficiency Flame-Retardant and Superhydrophobic Cotton Fabric by a Multi-Step Dipping"

_coatings, doi:10.3390/coatings11101147_

Round 1

Reviewer 1 Report

The authors describe the preparation method of cotton fabric with double coated layers. Using, the branched poly(ethylenimine) (BPEI) and ammonium polyphosphate (APP), and polydimethylsiloxane (PDMS)/cellulose nanocrystals (CNC)-SiO2 authors obtained a BPEI/APP/PDMS/CNC-SiO2(BAPC)composite coating on the surface of the cotton fabric. The resulting cotton fabric has been well tested using a variety of techniques: SEM. TGA, FTIR, XPS, mapping analysis. The method of cotton fiber modification used have a positive result. Prepared cotton fabrics have excellent flame-retardant superhydrophobic properties, and a good abrasion resistance. In my opinion, the article presents a simple, innovative and effective method of modifying cotton fabrics in order to give them the desired properties, and it is suitable for publication in Coatings.

Author Response

Response: Thanks for the nice comments.

Reviewer 2 Report

The paper describes the preparation of flame-retardant and superhydrophobic cotton textile obtained by multiple coating consisting of BPEI, APP, PDMS and CNC-SiO2. Manuscript is well written and structured, however it should be improved replying to the following comments:

  • In the text, some errors and inaccuracies are present, see in line 30, 52, 59, 142, 172, 265 and 266. Please check and correct.
  • The composition of coating, as described in Section 3.1, corresponds to an intumescent coating. Some comments about this and the references (Malucelli et al., RSC Adv., 2014, 4, 46024; Laufer et al., Biomacromolecules 2012, 13, 2843−2848; http://dx.doi.org/10.1016/j.tca.2014.06.020; Ortelli et al., DOI: 10.1016/j.jcis.2019.03.055) should be added.
  • XPS analysis has been used by authors for just providing identification of element present in the coating. XPS measurements usually is exploited for more sophisticated characterization. A quantitative analysis of surface coating by XPS should be provided.
  • BAC-cotton and BAPC-cotton show self- extinguishing properties. Some comments about this and the references (https://doi.org/10.1007/s10570-018-1745-z; Bosco et al., Surface & Coatings Technology 272 (2015) 86–95) should be added.

Author Response

Comments:

Dear author, it is recommended that you revise the paper from the following aspects.

  1. In the text, some errors and inaccuracies are present, see in line 30, 52, 59, 142, 172, 265 and 266. Please check and correct.

Response: Thanks for the good advice. The correction is as following, the line numbers have changed in the resvied manuscript.

Line 31: losses of human lives and properties

Line 53: the social demand for high value-added products

Line 60-61: a multi-step immersing. And a self-healing and flame-retardant superhydrophobic surface can also

Line 145: CNC-SiO2 rods

Line 172-173: CNC-SiO2 rods; (f) BAPC-cotton

Line 258-259: the pure cotton could be quickly lit and the fire spreads fast. And when the flame is re-moved, the fire does not go out until it is absolutely burned.

  1. The composition of coating, as described in Section 3.1, corresponds to an intumescent coating. Some comments about this and the references (Malucelli et al., RSC Adv., 2014, 4, 46024; Laufer et al., Biomacromolecules 2012, 13, 2843−2848; http://dx.doi.org/10.1016/j.tca.2014.06.020; Ortelli et al., DOI: 10.1016/j.jcis.2019.03.055) should be added.

Response: Thanks for the good advice. All the above papers have been added to the ref. 23-27 in the revised

manuscript. As following,

  1. Malucelli G, Bosco F, Alongi J, et al. Biomacromolecules as novel green flame retardant systems for textiles: an overview[J]. RSC Advances, 2014, 4.
  2. Alongi J, Cuttica F, Blasio A D, et al. Intumescent features of nucleic acids and proteins[J]. Thermochimica Acta, 2014, 591, 31-39.
  3. Laufer G, Kirkland C, Morgan A B, et al. Intumescent multilayer nanocoating, made with renewable polyelectrolytes, for flame-retardant cotton[J]. Biomacromolecules, 2012, 13(9), 2843-2848.
  4. So A, Gm B, Mb A, et al. NanoTiO 2 @DNA complex: a novel eco, durable, fire-retardant design strategy for cotton textiles[J]. Journal of Colloid and Interface Science, 2019, 546, 174-183.

  1. XPS analysis has been used by authors for just providing identification of element present in the coating. XPS measurements usually is exploited for more sophisticated characterization. A quantitative analysis of surface coating by XPS should be provided.

Response: Thanks for the good advice. We strongly agree that XPS measurements usually is exploited for more sophisticated characterization. But in our study, there are no a lot of chemistry going on and fewer new chemical structures to analyze. So, we think it should be also acceptable and to just provide identification of element present in the coating by XPS and this has served our purpose. There are some similar articles where only elements are analyzed by XPS, as follows:

Si Y, Guo Z, Liu W. A robust epoxy resins@ stearic acid-Mg (OH) 2 micronanosheet superhydrophobic omnipotent protective coating for real-life applications[J]. ACS applied materials & interfaces, 2016, 8(25): 16511-16520.

Xue F, Jia D, Li Y, et al. Facile preparation of a mechanically robust superhydrophobic acrylic polyurethane coating[J]. Journal of Materials Chemistry A, 2015, 3(26): 13856-13863.

   Considering that both N and P have not appeared in the BAPC-cotton of the XPS survey spectra (Figure 6), which may be because the CNC-SiO2 layer was too thick and BA was covered beyond XPS detection range. This has been described in article. But both EDS and FTIR results have been sufficient to describe the chemical composition and structure of the coating, in order to avoid confusing readers, we have decided to delete the "XPS Analysis" section.

  1. BAC-cotton and BAPC-cotton show self- extinguishing properties. Some comments about this and the references (https://doi.org/10.1007/s10570-018-1745-z; Bosco et al., Surface & Coatings Technology 272 (2015) 86–95) should be added.

Response: Thanks for the good advice. All the above papers have been added to the ref. 35-36 in the revised

manuscript. As following,

  1. Ortelli S, Malucelli G, F Cuttica, et al. Coatings made of proteins adsorbed on TiO2 nanoparticles: a new flame retardant approach for cotton fabrics[J]. Cellulose, 2018, 25(4), 2755-2765.
  2. Zhang M, Wang C. Fabrication of cotton fabric with superhydrophobicity and flame retardancy[J]. Carbohydrate Polymers, 2013, 96, 396-402.

Reviewer 3 Report

The paper presents the extensive characterisation of cotton fabric with flame-retardant and superhydrophobic properties and after different stages of immersing.  In my opinion it is interesting, well-structured, and the necessary information for repeating the coating procedures is presented. However, the authors are encouraged to proofread the manuscript, due to the presence of misprints and incomplete sentences. My recommendation is to publish the manuscript after minor revision. Please find below my suggestions:

  • Please clarify, why the BAPC-cotton showed N and P by the EDS mapping (Figure 5), and none by the XPS analysis (Figure 6).
  • In conclusion, the “good wear resistance” and “good performance” should be supported with numbers.
  • Since the motivation behind the work aims for a wider application, the discussion should be provided regarding the ecological aspects of the resulting cotton fabric, as well as the toxicity of the used materials (especially FOTS).
  • Too frequent use of word “successful”. My suggestion would be to make the paper sound more neutral.
  • Examples of detected misprints: line 71 (problems), line 172 (Fig. 1a), line 187 (residual mass), line 258 (2.3.7.), etc.

Examples of incomplete sentences: line 99 (Similar to the description in this study [19].), 

Author Response

Comments:

This paper can be published after following major amendment:

        1. Please clarify, why the BAPC-cotton showed N and P by the EDS mapping (Figure 5), and none by the XPS analysis (Figure 6).

Response: Thanks for the good advice. We think that Both N and P have not appeared in the BAPC-cotton, which may be because the CNC-SiO2 layer was too thick and BA was covered beyond XPS detection range. This has been described in article. But both EDS and FTIR results have been sufficient to describe the chemical composition and structure of the coating, in order to avoid confusing readers, we have decided to delete the "XPS Analysis" section.

  1. In conclusion, the “good wear resistance” and “good performance” should be supported with numbers.

Response: Thanks for the good advice. In conclusions, some data has been added supported with numbers. As following,

The flame-retardant and superhydrophobic cotton fabric could be prepared by a simple sequential dopping. The flame-retardant component BPEI/APP and the super-hydrophobic component CNC-SiO2 rods could be connected together by PDMS to form a multi-functional coating on the surface of cotton fabric. This as-prepared cotton fab-ric showed excellent flame-retardant and superhydrophobic properties, the value of oxygen index meter (LOI) reaching 69.8 and WCA is up to 156.6°. In the BPEI/APP coating, BPEI acts as acid source and a binder, APP acts as carbon source and a foaming agent. During burning, a loose carbon layer is formed on the surface of the cotton fibers to isolate heat and oxygen and to play a flame-retardant role. PDMS mainly adheres the CNC-SiO2 layer to the surface of BPEI/APP layer. In the CNC-SiO2 layer, the surface of SiO2 is hydrophobic modified by FOTS to provide rough structure and low surface free energy, while CNC mainly plays a role in stabilizing SiO2 particles; meanwhile, SiO2 is a highly effective flame retardant material. The BAPC-cotton fabric also shows good wear resistance. After the 16th abrasion cycles, the sample lost its superhydrophobic properties, but still had flame retardant properties. Given the simplicity of preparation and good performance, the resulting cotton fabric has a potential application.

  1. Since the motivation behind the work aims for a wider application, the discussion should be provided regarding the ecological aspects of the resulting cotton fabric, as well as the toxicity of the used materials (especially FOTS).

Response: Thanks for the good advice. The discussion has been added in lines 88-90. As following,

In this study, the used substances such as BPEI, APP, PDMS, and CNC-SiO2 are considered non-toxic; and the C-F bond is relatively stable in FOTS. However, they should be used with caution.

  1. Too frequent use of word “successful”. My suggestion would be to make the paper sound more neutral.

Response: Thanks for the good advice. The word “successfully” has been deleted in some sentences.

  1. Examples of detected misprints: line 71 (problems), line 172 (Fig. 1a), line 187 (residual mass), line 258 (2.3.7.), etc.

Response: Thanks for the good advice. These errors have been corrected.

Round 2

Reviewer 2 Report

The paper has been improved and  can be accepted for the pubblication.